# Making Capital of 'Illegal' Publication under Japanese Imperial Censorship: Publication Strategies of *Senki* (Battle Flag) around 1930

**Young Ran Ko [1,*], Nick Ogonek [2] and Kyeong-Hee Choi [2]**

1   Department of Japanese Language and Literature, Nihon University, Tokyo 156-0045, Japan
2   East Asian Languages and Civilizations, The University of Chicago, Chicago, IL 60637, USA;
    nogonek@uchicago.edu (N.O.); kchoi@uchicago.edu (K.-H.C.)
*   Correspondence: ko.youngran@nihon-u.ac.jp

**Abstract:** Around 1930, the Japanese publishing market was restructured, and as part of this process, the colonial market emerged within the Japanese Empire. In an attempt to expand into the colonial market, publishers such as Kaizō-sha, Chūōkōron-sha, and Senki-sha competed among each other, producing 'legal' and 'illegal' commodities related to socialism. This paper examines the circulation of illegal commodities such as the often-banned magazine *Senki* (Battle Flag), cross-reading them with internal documents from Senki-sha (*Senki*'s publisher) and NAPF (All-Japan Federation of Proletarian Arts), as well as with those from the Japanese Home Ministry and the Japanese Government-General in Korea. By doing so, the essay argues that the main actors of the socialist cultural movement around 1930 purposefully planned to capitalize on the 'illegal' nature of their commodities, while adopting a public stance of differentiation from commercial capital. Furthermore, by proposing that the publication of illegal commodities was in fact deeply imbricated with the movement of capital in the publishing market, this paper also reveals that Korean-language publications–notably, the magazine *Uri tongmu* (Our Comrades)–produced by socialists in the Japanese interior around 1930, ended up playing a role in undermining the reconstruction of socialism in Korea. For this reason, it is crucial to reconsider the prevailing narrative about the history of the Japanese socialist movement of the late 1920s and early 1930s, which often essentializes the connection between Japanese and Korean socialists as pure ideological solidarity, paying little attention to the complex movement of capital, legal and illegal, at work in the Japanese Empire around 1930.

**Keywords:** *Battle Flag*; censorship; Imperial Japan; colonial book market; proletarian literature

## 1. Add-On Value Brought by Publication Bans: *Senki* and "The Crab Cannery Ship"

The proletarian cultural movement began in 1921 with the release of the first issue of the literary magazine *Tanemakuhito* (The Sower). Another magazine called *Bungei sensen* (Literary Arts Front) became the center of the movement in 1924, and in 1925 the Japan Federation of Literary Arts was formed. This period–from the 1920s until the destruction of the Japan Federation of Proletarian Culture (KOPF) in 1934–represented the peak of the proletarian cultural movement in interwar Japan. One of the most preeminent magazines of this period was *Senki* (Battle Flag). *Senki* was launched in March 1928, when the Japan Proletarian Art League and the Union of Avant-garde Artists jointly formed the NAPF (All-Japan Federation of Proletarian Arts), merging their journals, *Puroretaria geijutsu* (Proletarian Art) and *Zen'ei* (Vanguard).

*Senki* was published until May 1930, when a police raid and mass arrests led to the dissolution of the editorial organization. Below is the circulation data of *Senki* until just before the mass arrests (Table 1).

**Table 1.** *Senki*: Issues Published and Issues Prohibited.[1]

|  | Issue Number | Number of Published Copies |  | Issue Number | Number of Published Copies |
|---|---|---|---|---|---|
| 1928 | First Issue | 7000 | 1929 | May | 12,000 |
|  | June * | 7300 |  | June * | 13,000 |
|  | July | 7000 |  | July | 13,000 |
|  | August | 6000 |  | August * | 14,000 |
|  | September | 7000 |  | September * | 15,000 |
|  | October | 8000 |  | October * | 16,000 |
|  | November * | 8000 |  | November | 17,000 |
|  | December * | 8000 |  | December | 18,000 |
| 1929 | January | 10,000 | 1930 | January | 20,000 |
|  | February * | 10,000 |  | February * | 21,000 |
|  | March | 10,000 |  | March * | 22,000 |
|  | April * | 11,500 |  | April | 22,000 |

* Indicates sale of the issue was prohibited by censors.

Norma Field and Heather Bowen-Struyk write that proletarian journals like *Senki* were dangerous to possess and as a result were passed around from reader to reader, meaning that "actual readership was even higher than circulation numbers indicate" (Field and Bowen-Struyk 2016, p. 4). And yet, the overall trend of *Senki's* intended and actual publication is roughly comparable to the circulation of *Chūō kōron* after 1927, when sales began to lag because of the growing popularity of the newly emerging cheap paperback book format (*enpon*) (*Shimbun zasshi-sha toku-hi chōsa* 1979).[2]

The September 1928 issue of *Chūō kōron* indicates, for instance, that the president had been replaced, and change in editorial direction was conducted by the newly appointed president, Shimanaka Yūsaku. Before Shimanaka's tenure, the magazine was an emblem of 'Taishō Democracy,' and published work from a number of major liberal intellectuals, including Yoshino Sakuzō; but Shimanaka employed many writers from the proletarian *Senki* in order to compete with *Kaizō*, which published many writers from *Bungei sensen*. In describing *Chūō kōron's* editorial strategy during the period, which aimed to repel the challenge of *Kaizō* and seize the initiative, then-editor Amamiya Yōzō stated that "The owner, Shimanaka, approached Marxist thought like it was in vogue, like it could be material for the magazine alongside cinema and sports," noting that it was at this time that proletarian writer Kuroshima Denji's story about the Siberian exodus, "Hyōga" ("Glacier"), planned for inclusion in *Chūō kōron's* special issue in January 1929, was banned and the magazine pulled from sale. In a memoir of this time, Amamiya writes:

> While reading and proofreading [Kuroshima's] novel with President Shimanaka, I told him that it might cause problems with the authorities, but he was bullish, and assured me there was nothing to worry about. When we heard that the issue was banned, he was impressed that I had such a keen eye, but I think we just had a different sense of the risks publishing faced at the time. Indeed, after this special New Year's issue was banned, Former President Asada seemed extremely worried about the economic impact, but Shimanaka believed that it would sell well once we had made the changes the censor required. Although it frayed the nerves of all involved, at the time, editorial work was nothing less than ideological and moral warfare, where defeat meant a publishing ban and victory meant release to the commercial market. So, even if the magazine was fundamentally liberal in orientation, a harder left-wing core formed among us editors. (Amamiya 1998, p. 536)

It should be noted here that in order to form the editorial left wing at *Chūō kōron*, and counter rival *Kaizō*, Amamiya was drawn to the increasingly successful *Senki*, whose circulation had risen to around 20,000, rather than *Bungei sensen*, whose circulation was only several thousand. Page space would also be devoted to the writers of *Bungei sensen* when they

were considered to have commercial value (Amamiya 1998, p.536). To the editors and publishers of *Chūō kōron*, who were trying to rebuild their editorial strategy and business along a leftist line, *Senki*'s increase in circulation to more than 20,000 copies was very attractive and seemed to prove the efficacy of the strategy. But with this increase in *Senki* circulation came an intensifying frequency of publication and distribution bans, as Table 1 shows. In conjunction with this data, the "socialist" competition between Chūōkōron-sha and Kaizō-sha, the publishers of *Kaizō*, the representative publishing capital of Imperial Japan became visible, leading to a spike in the number of publishers specializing in left-wing books (Ko 2009, 2010b; Wada 2022).[3] The terms "left-wing publishers" (sayoku shuppan-sha) and "leftish publishers" (sayoku-teki shuppan-sha) even appeared.[4] Left-wing publishers were "class-based publishers that printed and published books based on the activist tactics of the proletariat," while leftish publishers were "bourgeois publishers in form and leftish book publishers in content" because they utilized the capitalist distribution network (Kanroji 1931, p. 348). Kanroji Hachirō writes that the major leftish publishers were Kibōkaku, Dōjinsha, Kyōseikaku, Sōbunkaku, Marx Shobō, Iskura-kaku, Hakuyōsha, Sekaisha, Tetsutōshoin, Ueno Shoten, Nanban Shobō, Nansō Shoin, and Kōbundō. Hakuyōsha was the only "leftish publisher that cannot be trusted". The reason was simple: its books were never banned. As Kanroji points out, the main competitors of leftish publishers are the bourgeois publishers who are not afraid to commercialize socialism, rather than the left-wing publishers.

For example, there was the fierce competition (Umeda 1998)[5] between the 'leftish publishers' (an edition produced by a coalition led by Iwanami Shoten, and including Kibōkaku, Dōjinsha, Kōbundō, Sōbunkaku) and the 'bourgeois publishers' (Kaizō edition) over the publication of the Complete Works of Marx and Engels in 1928, recorded as the "greatest tragedy in the history of *enpon*" (Obi 2007, p. 298). In the end, the Iwanami-led version was defeated and the coalition dismantled, and not a single volume could be published. The enormous losses from the investment in translation, typesetting, advertising, etc., were borne mainly by Iwanami Shigeo, founder of the publisher Iwanami Shoten (Obi 2007; Cheon 2003)[6] When publication of the Kaizō edition began, the Special Higher Police, together with the Ministry of Education, conducted a thorough investigation of those who had applied to pre-order books, mainly students and teachers.[7] Based on this information, a blacklist of 'left-leaning elements' was compiled, and those on the list were explicitly excluded from being hired as elementary school teachers.[8] At a time when the *Yomiuri* newspaper reported a flurry of 'damage' caused by 'legal' publications circulating in the 'legal' market, where the yen production-distribution-consumption system of *enpon* operated, there was heightened demand for 'illegal' products whose very possession was deemed dangerous, by intellectual readers such as students and teachers.

There was not a rigid or stable distinction between left-wing and leftish publishers, however, because many publications changed their strategies throughout the period and many writers published work in both kinds of outlet. Indeed, the *Senki* coterie writers, who were regarded as the base of left-wing publishing, came into the limelight because they moved freely between bourgeois publishing, leftish publishing, and left-wing publishing. In particular, Nakano Shigeharu, who wrote the well-known poem alluding to the Emperor's assassination, "Ame no furu Shinagawa-eki" (Shinagawa Station in the Rain), and Kobayashi Takiji, who made his debut during this period, were rapidly emerging as talented and popular writers. Kobayashi's story "Kanikōsen" (The Crab Cannery Ship) was serialized in *Senki* in May and June 1929. Of these, the June issue was initially pulled from sale. As demonstrated in Table 1, *Senki* was frequently pulled from sale months starting in November 1928, and the banning of the June 1929 issue could have been easily predicted given the mass arrests and imprisonment of communists and labor activists in the infamous April 16 Incident, which took place during the editing and printing of the issue. But when the issue was released, containing the second half of "The Crab Cannery Ship," the newspaper *Tōkyō Asahi Shimbun* (Tadato Kurahara, "Works and Reviews", June 17, 1929) and literary journal *Shinchō* (Seiichiro Katsumoto, July 1929) published favorable reviews

of it, and the general readership responded positively (Ogasawara 1985; Shimamura 2008; Field 2009).[9]

The stricter the censorship against socialism became, the more manuscript requests were made to NAPF-affiliated writers, and the more revenue was generated from printing books by these authors. Amamiya noted that "the so-called bourgeois literary world was dismayed that *Kaizō* and *Chūō kōron* were competing so forcefully to publish works by professional writers" (Amamiya 1998, p. 537). But there was another aspect to this arrangement. Amamiya continues that "Kobayashi Takiji was so pleased by the results from my negotiations to publish his work in *Chūō kōron* that he danced for joy," and when "Kobayashi and his colleague, the equally great proletarian writer Tokunaga Sunao, were too thrilled to speak when they came to collect the manuscript fee, and thanked me later by letter instead" (Amamiya 1998, p. 539). Of course, it is necessary to take Amamiya's reflections–recorded so long after the fact–with a grain of salt. However, they speak to the atmosphere that enabled NAPF to embark on a publishing business, and demonstrate its opposition to censorship in order to create a unique brand as the 'headquarters of left-wing publishing'.

When publishing "The Crab Cannery Ship" in book form, Senki-sha restored all of the *fuseji* that had appeared in its original publication in the magazine *Senki* (Toda 2019).[10] *Fuseji* are censorship marks that made the traces of deletion visible. In this period, editors often 'voluntarily' used them to cover expressions that might violate regulations or otherwise put them at risk from authorities. In this way, contrary to the original function of *fuseji*, to conceal, they functioned as a formal device to encourage reading comprehension of the concealed material (Maki 2014, p. 15). In this sense, while *fuseji* were a symbol of submission to censorship and the resulting humiliation, they were also a form of resistance to censorship (Yamamoto 1967; Ko 2006, 2010a; Abel 2012; Maki 2014).[11] Senki-sha's publishing strategy is clearly shown in the text used in a series of advertisements that appeared in *Senki*.

> **September 1929 issue:** Prepare for the book's inevitable ban! Always pre-order! (Best timing: just before it goes on sale)
>
> **October 1929 issue:** Defend proletarian publications against censorship! Order directly!
>
> **November 1929 issue (Figure 1):** The first edition was immediately prohibited from sale.
>
> The first edition was sold out before publication, and a revised edition with a new binding will be issued.
>
> Protect class-based publications from bans! You can be sure of obtaining a copy if you pre-order in person!
>
> **December 1929 issue:** More editions on the way! The sixth edition is published with the overwhelming support of workers and peasants throughout the country!

The first edition of *The Crab Cannery Ship*, which used few fuseji, was banned on its day of publication. In this edition, the novel was published together with another Kobayashi story, the more closely regulated "15 March 1928", about the March 15 Incident, a mass-arrest of leftists. As shown in the text of Figure 1, this story was removed in its entirety from the revised edition. However, the revised edition published in accordance with the stipulations described in advertisements (with some phrases covered with *fuseji*) was also immediately banned. In March 1930, a revised and popularized edition was published with more *fuseji*, as well as "full-text *rubi*," or phonetic kana over the kanji, making the text much more friendly to people with low literacy skills. The advertisement in Figure 2 reports that a total of 16,000 copies were issued. The successive bans attracted so much attention that there was a rush of orders; so many, in fact, that there were conflicts among the distributors over how copies would be allocated. The following testimonies from 1931

show how sales-sensitive booksellers reacted to the publication of the book *The Crab Cannery Ship*:

> It didn't matter whether the price is low or high. I was told by a Tokyo bookstore that when *The Crab Cannery Ship* first came out, they thought it was going to be banned, so they kept it in the back of their shop, in storage, instead of selling it. This is because there was a sense that such a book will inevitably be able to go back on sale (Kanroji 1931, p. 244).

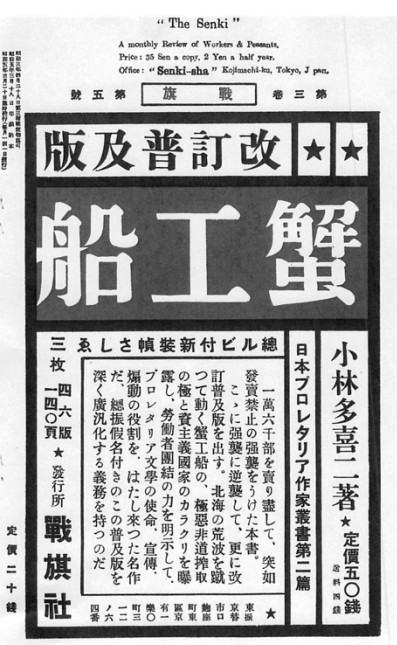

**Figure 1.** Advertisement for *The Crab Cannery Ship*, *Senki,* November 1929.

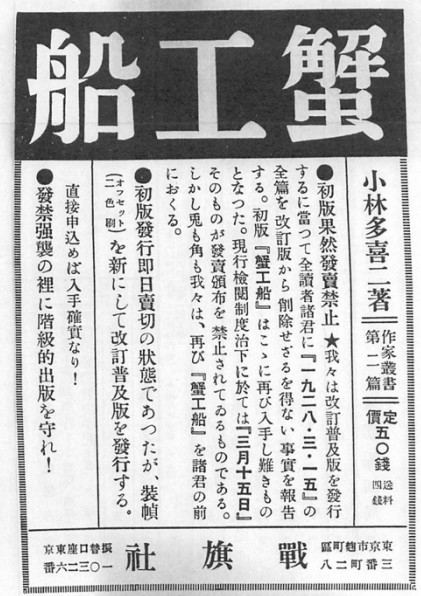

**Figure 2.** Advertisement for *The Crab Cannery Ship*, *Senki,* March 1930.

The strategy of the newly formed publishing house, Senki-sha, was to visualize its opposition to censorship and government power; in other words, to acquire the added value of being banned, and capitalize on the 'illegal' label. It was a movement for capital acquisition, not only an ideological movement. Therefore, Senki-sha's position cannot

be ascertained by merely foregrounding the damage of severe censorship to its medium and the ideas inscribed therein. Indeed, although *Senki* was closely surveilled and aggressively censored, it remained nonetheless highly culturally and politically influential. In the remainder of this essay, I will focus on the distribution of socialist magazines and books that moved between the boundaries of 'legal' and 'illegal' definitions set by the Imperial power in the 1930s. I aim to consider the term 'illegal' and the process of capital generation related to it. This capital generation abounded in the pages of magazines such as *Senki*, which both the publishing police and the "Special Higher Police" or Tokkō, classified as ultra-left wing and subjected to special vigilance. I will also discuss how this period coincided with the discovery of the colonial publishing market by imperial publishing capital and the commodification of socialism, and explore how leftist publishing engaged in this restructuring of the publishing market.[12]

### 2. *Senki* and Capitalizing on 'Illegal' Goods

Amidst increasingly punitive and extensive censorship, how was Senki-sha able to ensure the mass distribution of banned books, with customers even openly purchasing these 'illegal' commodities through direct order? Was it the incompetence of the Imperial authorities, or did Senki-sha alone find some way to avoid the pressures of the times, which included significant factors such as the amendment of the Peace Preservation Law, and the subsequent March 15 Incident and April 16 Incident?[13] Unlike the March 15 incident, when the general press was able to stage a performance-like media event (Okudaira 2006, p. 110), by competitively reporting the mass arrest of 3400 people nationwide, the April 16 incident in 1929 was a pinpoint attack, recorded as an incident that demonstrated the expanded influence and power of the Tokkō (Ogino 1984, p. 213). In the wake of the March 15 incident, at the beginning of April, a fund of 2 million yen was allocated for the expansion of the Police Affairs Bureau, mainly for the Special Higher Police Department and the Book Department. The sudden expansion of the budget was not a request from the administrative level of the Home Ministry, but rather a directive from "higher up," (Ogino 1984, p. 171) and those in charge had difficulty fully executing the newly swollen budget. For example, Tsuchiya Shōzō, then the head of the Book Department, said, "The Director [of the Police Affairs Bureau] called me and told me to double the budget of the Book Department, but it was not easy. […] Ultimately, we doubled the budget by increasing the number of clerks and censors on the payroll, and by devising and publishing circulars and other internal reports" (Tsuchiya 1967, p. 50).

Through this process, the *Shuppan keisatsu hō* (Publication Police Reports) were published and the number of personnel in the Book Department was expanded from 24 in 1927 to 61 in 1929. In addition, the monthly internal publications, detailing censorship activities, procedures and policies were also launched to help Tokkō chew through its budget. Other materials were compiled six times a year for the education of Tokkō officials, and various reference books were also published. The content of the magazines and books published by the Book Department and the Tokkō with their expanded budget was largely an analysis of trends in arrests of socialists and censorship statistics. The sharp increase in the number of special investigators at this time shows the same curve as Table 1, on the rapid growth in *Senki* sales; in October 1927, there were 4401 special inspectors, which almost doubled to 8043 in October 1929 (Shakai undō no jōkyō 1971).[14] This was also due to the increased budget of the Police Affairs Bureau, and the sharp increase in the number of bans-from 216 cases in 1927 to 935 cases in 1929-under the designation of the Publication Law and the Law of the Temporary Control of Seditious Documents, especially those related to the 'public peace and order' (Yui 1985, p. 58).

Along with the increase in the budget, surveillance of Koreans was also expanded. At the direction of the Home Ministry, the Tokkō began to maintain a list of all the approximately 20,000 Koreans living in Japan, and once or twice a month conducted door-to-door check-ins and nighttime house-to-house searches. Furthermore, in order to prevent the movement of Koreans into the interior of Japan, the Ministry of Foreign Affairs also fo-

cused its efforts on countermeasures against 'malcontent' Koreans, such as mobile police units, coastal patrols, and language classes. The stated reason was that "overseas Korean malcontent groups are plotting with left-leaning Koreans in Japan's interior to accomplish their anti-social plans in anticipation of the Imperial Enthronement Ceremony."[15] The discourse of the ruling power that initiated the March 15 and April 16 incidents linked 'socialism' and 'Korean malcontents' through the medium of expressions such as 'disrespect for the Emperor' and 'high treason'. In the September 1928 issue of *Senki*, both "Through the Steel Bars" (by anonymous Taka[XX] Tarō in Nagoya Prison) and "Deported" by Yi Puk-man appeared under the issue's special theme of "Deported and Incarcerated Comrades" in relation to the March 15 Incident. In his article, Yi Puk-man asks, "What 'crime' have I committed that they would persecute me so relentlessly?" Thus, *Senki* was a venue where 'public enemies' as defined by the Tanaka cabinet at the time came into contact with each other.

Tsuboi Shigeji, who worked in *Senki's* management department and was directly involved in financial matters during the magazine's period of rapid growth, gave the following account: "*Senki* is banned from publication every month. Not only that, it is also banned from being released before it is published and delivered to distributors, and the reality is that the censor's minions take all the copies. If *Senki* were a bourgeois magazine that only pursued profit, we would simply cease publication because there is no opportunity to make money". He continued that, for proletarian magazines, "Bans and seizures are fatal, rather than a lack of profit. […] [T]herefore, even if bourgeois laws banned the release of the magazine, we devised methods to prevent it from being seized by the authorities, and employed management methods particular to proletarian magazines for the magazine's economic defense."[16] Tsuboi revealed that Senki-sha utilized two distribution methods, a 'bourgeois' distribution network (through book shops) and a direct distribution network (though local *Senki* branch offices).

A brief article entitled "Appeal to All Readers" published in *Senki*, shows that in the early years of the magazine's existence, when it was still a legally published magazine, it actively sought to utilize the 'bourgeois' distribution network. This Appeal, from the September 1928 issue, reported bans on the magazine's release, the detention and arrest of NAPF officials who published it, and the forced search of NAPF headquarters on August 7th, among other things. The Appeal concluded, "If you find any bookstores that do not stock this magazine, please ask them to order it from the major sales outlets as soon as possible. Posters advertising the magazine should also be placed in bookstores and other locations throughout the country."[17]

In order to solve the problem of increased magazine production costs and defray the risk associated with publishing a left-wing magazine, Senki-sha borrowed (1) The distribution and consumption channels of 'bourgeois publications' that used large distributors (such as Tōkyō-dō) and general-interest bookstores, (2) The distribution channels of leftish publications that operated their own membership system, and (3) Direct distribution through local *Senki* offices and reading groups, while repeatedly emphasizing the challenges of censorship, suppression and publication bans in the magazine's pages to drum up support and establish their brand. As can be seen in Tsuboi's statement above, *Senki* owed its success in part to the added value of it being banned, and a unique distribution network was devised based on (3) direct distribution, distinct from the production, distribution and consumption system of (1) bourgeois publishing and publishing capital.

Incidentally, connecting the distribution scheme with the question of content reveals an interesting structure. First, let us note that *Senki* was launched in 1928, at the peak of the *enpon* boom, and that until early 1930, when the magazine experienced dramatic growth, print capitalists were engaged in a fierce competition to capitalize on 'socialist' commodities. Who provided the content for the 'socialist' commodities that fed the 'bourgeois' press? An answer to this question occurred in 1930–1931. The anthology *Senki Sanjyūroku-jin shū* (Thirty-Six War Flags 1930), which contained many 'illegal' pieces, appeared on the publishing market as a collaboration between Kaizō-sha and Senki-sha (Eguchi and Yamaji

1930). The reason why the NAPF entered a joint venture with the Kaizō-sha rather than just using its own publishing organization, Senki-sha, was outlined in the March 1931 issue of the magazine *Nappu* (the theoretical organ of the NAPF, published by Senki-sha).

> Kaizō-sha co-published the anthology *Thirty-Six War Flags*. […] Major works which would have served to represent each author were not included, (because permission to publish the works published by Senki-sha was not granted). The book was edited to cover the author's most recent works. We can assure readers that each of the thirty-six pieces here are representative of their authors' best work. […] The reader might wonder what the purpose of publishing this anthology was. As Eguchi has stated in his preface, there were many class casualties among our comrades in 1930. This publication is to provide material support to their families. All royalties will be sent to their needy families.
>
> Kiji Yamaji, "Introducing the New Publication *Thirty-Six War Flags*" (Eguchi and Yamaji 1930, p. 121)

As mentioned previously, high-profile members of the NAPF, including Kobayashi Takiji, Nakano Shigeharu, Hayashi Fusao, Tsuboi Shigeji, and Murayama Tomoyoshi, were imprisoned for more than a year following the mass arrests of Senki-sha workers and collaborators in May 1930. Many of the writers listed in *Thirty-Six War Flags* were also included there. As it is emphasized that only their representative works were published by *Senki* itself, the *Japanese Proletarian Writers' Series* published by Senki-sha the same year included "Kanikōsen" (The Crab Cannery Ship, by Kobayashi Takiji), "Tetsu no hanashi" (The Story of Tetsu, by Nakano Shigeharu), and "Taiyō no nai machi" (The Town Without Sun, by Tokunaga Sunao), among others, as *Senki's* representative works. These books were "representative of the *Senki* authors, and did not often appear in remainder sales at the night market in Jinbōchō [a major book selling neighborhood in east Tōkyō], because they have sold a considerable number of copies already and should continue to sell well" (Kanroji 1931, p. 240).

Although Senki-sha's advertisement for the Kaizō-sha co-produced *Thirty-Six War Flags* is explicit that it does not contain "representative" works by *Senki* writers, the inclusion of such works in Senki-sha's own book series is particularly emphasized. It thus becomes clear that works by the same authors were simultaneously marketed through the distribution routes of 'bourgeois publishing' and 'left-wing publishing'. After the 1930 Senki-sha incident, it became impossible to sell the company's publications in bookstores. Therefore, they attempted to use the legal distribution system to generate income from royalties, although there was significant internal protest against such a move. For example, the editors even adopted a resolution that *Senki* should increase its circulation figures "in order to attract readers to proletarian publications," because "thanks to their inclusion of work of proletarian writers, bourgeois magazines' sales figures already bolster their sales figures by ensnaring readers who originally sought proletarian publications."[18]

From the very beginning of its publication, *Senki* set a goal of securing 20,000 readers. However, while attempting to diversify their risk in various ways, they began to aim for the same one million readers of the magazine *Kingu* (King, published by Kōdansha). The "mass = 1 million readers" calculation had already appeared in debates about the uses of popular art between Kurahara Korehito and Nakano Shigeharu, two of NAPF's leading theorists, which had the effect of raising the profile of the first issue of *Senki* (Shockey 2019).[19] Interestingly, despite the fact that both sides of the debate could not resolve their opposing views, their specific goals for popularization in the debate were identical.

While analyzing this debate, Maeda Ai focuses on the emergence of the concept of 'masses' and discusses the discourse which posited the 'masses' being 'enlightened' by *enpon* (inexpensive paperbacks of classic literature, foreign literature and philosophy) and Kōdansha culture. In short: "The new challenge of how to win the 'masses' back to the side of proletarian literature and politically 'enlighten' them was faced by the proletarian literary movement" (Maeda 1989, p. 208).

Furthermore, Satō Takumi, scholar of *Kingu* magazine, analyzed the efforts of the *Senki* group to win readers as a "counter-movement that directly challenged the mass public nature of *Kingu*." (Satō 2002; Perry 2014)[20] A common point among the discussions of the popularization movement is that they consider the establishment of print capitalism in the same period as chiefly a problem of capital, while the 'head-on opposition movement' by the *Senki* group and others is considered only as a problem of the agitprop that "opposed bloated bourgeois journalism," far removed from capital formation (Maeda 1989, p. 209). From such a perspective, it is impossible to read into the meaning of the statement by Tokunaga Sunao, a star writer produced by *Senki*, that "Our *Senki* can achieve its economic objectives only by recapturing the common workers who are the readership of *Kingu*" (Tokunaga 1930, p. 244).

To emphasize once again, *Senki* appeared during the heyday of *enpon*. Through applying the advertising, publicity, and distribution system of the mass-media magazine *Kingu*, *enpon* created a huge publishing phenomenon thanks to mass book production, low-cost sales policies, loyalty programs, advertising with a radical and commanding tone, lectures by famous authors for readers, and so on. *Senki* was also trying to access this market and its capital using similar strategies to *enpon* publishers. In other words, *Senki* was not immune to the sphere of influence of print capitalism, which was booming at the time.

While differences in the use of capital acquired in the publishing market should be taken into account, as will be discussed in the next section, the bottom line is that the structural characteristics of the *Senki* and the Senki-sha book editions, i.e., the publication competition and struggle of the treasonous *Senki* coterie (NAPF) that advocated overthrowing the Emperor System (*Kingu*) since the 1927 Thesis, were not a purely ideological movement that developed outside the existing print capital. It is important to be aware that they, too, were internal to the system of print capitalism and, like other commercial magazines such as *Kaizō*, formed an axis of the capital movement that dreamed of recapturing the masses, the readers of *Kingu*.

Of course, the magazine *Senki* and the publisher Senki-sha also produced books through collaboration with paper producers, printers and bookbinders, and conducted business not only with book distributors and bookstores, but also with workers and farmers who were members of their direct distribution network through the medium of yen. Their agitprop activities were based on the collection of this yen; the collection, in other words, of capital.

### 3. *Senki* as a Catalog of 'Illegal' Commodities

Around the end of 1928 (November 1928, December 1928, and February 1929 issues), when *Senki* was continually pulled from circulation, the marketing mottos "Become a reader yourself" and "Expand the reader distribution network throughout the nation" appeared repeatedly in the magazine's pages. The 'nation,' included Imperial Japan's colonies such as "Chōsen [Korea], Taiwan, and other areas," where "Senki's publication was XX [*fuseji* concealing the word 'banned'] before it is issued every month" (March 1929). The word 'Chōsen' appeared in a variety of ways in *Senki*. From the first issue (May 1928), three articles by Yi Puk-man ("The Past and Present of the Korean Proletarian Movement (2)," "On the Occasion of Welcoming May Day," and "Deported") and "Our Day Is Near" by Yi Byung-ch'an appeared. In addition, "The Truth of the Clash in Kawasaki" (July 1929), which dealt with an armed clash between Korean workers in Kawasaki, an industrial center outside Tōkyō, was written by Kim Tu-yong, the Chair of the Central Executive Committee of the Federation of Trade Unions of Zainichi Koreans in Japan. From the end of 1929, he took the lead in dismantling the Korean workers' organizations on the mainland and organizing them under the umbrella of the Japan Trade Union National Council (or Zenkyō).[21] The structure of *Senki* (such as the placement of the table of contents) and the information conveyed by Yi Puk-man and Kim Tu-yong in their writings were perceived as representing the voices of "comrades and compatriots" from "the whole country and the Korean provinces" of Japan. For example, "Floods in Korea: Help Our Korean Brothers in Disaster!" (Pak Wang-yang, September 1928), denounced contemporary relief movements,

asserting they were impossible inside Korea, because of the prohibition of relief donations in the affected area, Kunsan, and urged "Dear Japanese Comrades," to "send relief funds" to directly their "comrades in white coats!" The letter contained instructions to address all remittances to Senki-sha.

One of the reasons for the Tokkō's intensive pursuit of *Senki*'s readership organization and network was to reveal the relationship between the Japanese Communist Party (JCP) and the NAPF.[22] *Senki* published statements in support of the JCP, and its offices were used as operations and editorial spaces for *the Second Musansha Newspaper* (legal from December 1929), the *Agricultural Labor League* and the *Musansha Youth Newspaper*; all published by a Senki-sha bureau dedicated to rebuilding the JCP. It also became problematic because it was acting as a proxy for the solicitation of funds for organizations that had come to the attention of the Tokkō, such as the Defense Fund for the *Musansha Newspaper* and the Liberation Movement Victims Relief Society.

During this time, *Senki* was a kind of catalog for various 'illegal' goods and 'illegal' fund solicitation. All purchases required payment in advance. When the Liberation Movement Victims Relief Society sold hand towels (advertised in the August 1928 issue, 15 sen per towel plus postage), the magazine announced, "We have received inquiries from Taiwan, Korea, Manchuria, and even the U.S., but we are still unable to ship many orders because they sold out so quickly" (September 1928). 'Lenin' was also one of the magazine's main commodities. For example, a bust of Lenin was introduced as a fundraising item in the same issue for 75 sen. The same bust was also advertised in the *Musansha Newspaper* Club for 80 sen. Large portraits of Lenin ("Brighten up your rooms, comrades, and support the relief society!" August 1929 issue) and other items frequently appeared as fundraising items. Senki-sha itself sometimes developed commodities and sold them. Among these were a commemorative photograph of the Workers' and Peasants' Party Founding Convention (February 1929 issue, 20 sen per copy plus postage), which was sold to mark the fact that the audience protested after speeches were canceled by the Tokkō, and other moments where 'illegality' was staged. The advertisement for the photo sale was placed at the end of the article reporting on the convention, and above the advertisement was written, "A living example of the dialectical struggle to attain legal status by using and subverting 'legality'". In other words, the boundary between 'illegality' and 'legality' was not clear, and the oppressed could strategically stage their 'illegality'.

Above all, Senki-sha was engaged in securing direct funding. In February 1929, as *Senki* was hit with recurring sales bans, the magazine published an article titled "How to Organize Local *Senki* Branch Offices!" in which the company presented a concrete proposal for the establishment of branch offices and the formation of a readers' association. The price for purchasing *Senki* directly, at a bookstore, or through a readers' group or *Senki* branch office would be 35 sen per copy. The more times *Senki* was banned, the more emphasis was placed on the phrase, "If the readers of bourgeois magazines are mere consumers of commodities, the only source of revenue for proletarian magazines is the magazine fee paid in advance to the branch offices" (Tsuboi 1930, p. 198).

In May 1930, the Senki-sha mass arrests led to the seizure of the branch office directory and the complete destruction of the organizational network the publisher constructed. Until this point, the Tokkō had no grasp of the entire organization of Senki-sha, and had to rely on publicly available information to conduct its investigations. Immediately prior to the mass arrests, the *Tokkō Geppō* stated, "The producers of *Senki* are becoming more and more cunning day by day. From April, the publishers proclaim the necessity of a fund of 3000 yen. This demand seems to be having the desired effect."[23] This indicates that the branch offices and reading groups were systematically mobilized for the fund solicitation by *Senki*. In other words, the reason why the Tokkō monitored the branch office of *Senki* was not simply because it generated 2600 yen in revenue; it was also because the 20,000 readers of *Senki* had become material supporters of this 'illegal' organization, funding it through round-about channels such as by contributing to the fund to relaunch the *Musansha Newspaper* (Japanese Communist Party Reconstruction Movement) and the fund of

the Liberation Movement Victims Relief Society (which supported detained comrades and their families).

An increase in *Senki's* sales meant an increase in the number of branch offices, and an increase in branches meant an increase in the number of 'mass' readers, including workers, peasants, and others. The names of purchasers of all 'illegal commodities' appearing in the pages of *Senki* include many Koreans. In addition, a few unknown Koreans of a different class from Kim Tu-yong and Yi Puk-man, who conveyed socialist theory and the situation of the struggle in the interior in Japanese, appear in *Senki*, although in small numbers. In the "Readers' Room" section (January 1930 issue), a Korean worker wrote that it took him "two and a half days" to write a single postcard because "I am not a learned man". This Korean, who said he had no money and could not go to his mother's funeral, concluded his missive with determination, "I will defend *Senki* even if I lose my head". In those days, 'defending' *Senki* meant nothing more than expressing one's intention to send funds.[24]

Immediately after being banned seven times over the course of a year, the editorial board of *Senki* published "On the Writing of Letters–An Appeal to All Readers" (November 1929). It is a reminder of the lack of efforts towards popularizing the magazine up to this point, and an invitation for more submissions from readers. The number of pages devoted to readers' comments increased from the end of 1929 to the beginning of 1930. Many readers began to request the addition of *rubi* pronunciation marks and the elimination of difficult characters and expressions. For example, "'The Crab Cannery Ship' is a powerful work, but it is a little complicated. I think it would be better if it were simpler. We couldn't quite follow it". (letter from the Kakuda Printing Company Youth Laborer Group, July 1929). Later, Kobayashi Takiji, the author of the story, published "Ginkō no hanashi" (Bank Story, April 1930 issue), a very simple narrative with almost no kanji characters. Thus, in order to withstand the financial hit that the sales ban caused, which became more severe around 1930, *Senki's* strategy of popularization was accelerated, and its sales circulation thus exceeded 20,000 copies. Then, in early 1930, *Senki* the mass-audience magazine was born in the process of staging a joint struggle with the branch offices and reader-comrades over the sales ban, albeit for a short period of time. The aforementioned top theorists of the *Senki* school (see Nakano and Kurahara's debate on popularization) were unable to present an alternative to this strategy. In the end, the greatest cause of *Senki's* evolution into a mass-audience magazine was the sales ban and the continued suppression by the Tokkō.

### 4. *Uri tongmu* (Our Comrades) and the Colonial Book Market

At the end of 1931, when the pressures faced by the *Senki* coterie were becoming increasingly serious, the Japan Proletarian Cultural Federation (KOPF) was organized, and its journal *Puroretaria bunka* (Proletarian Culture) was launched. The KOPF launched *Hataraku fujin* (Working Woman) in January 1932, *Taishū no tomo* (The People's Friend) in February 1932, and *Chīsai dōshi* (Small Comrades) for children in March 1932. Specific left-wing magazines targeted different audiences, such as women, workers, and children; and sales methods and reader organizations of these magazines followed those of *Senki*.

The Tokkō paid close attention not only to the how funding for Senki-sha and the NAPF was collected and managed, but also to their sources. Even in its heyday in the 1930s, *Senki* was in poor financial condition, and the NAPF was aware of this.[25] Of particular interest to the Tokkō was the role of *Senki* and the NAPF as a source of funds for the JCP as an extra-Party organization. According to reports from the Tokkō, communication between the Party and the Comintern had been cut off since the April 16 Incident. As a result, the Party needed a financial boost for its campaigns, and "those who were sympathetic were divided into groups such as the NAPF, doctors, students, laborers, journalists, and so on, and representatives from each group were asked to solicit contributions". Those in charge of solicitation were told that "True comrades would be told that the money was for Party activities, while others would be told they would be donating to relief funds, newspaper funds and so on."[26]



Many of the items published in and sold through *Senki* were related to the relief funds and *Mushin* (a newspaper for the unemployed). More than half of the income recorded in the "Survey of Income from the JCP Activity Fund" from July 1929 to January 1930, prepared by the Tokkō, was secured through the NAPF (from August 1929 to January 1930).[27] Significantly, the increase in donations to the NAPF was proportional to the increase in sales fees for *Senki*. Therefore, the Tokkō was more interested in the funds raised through the 'joint struggle' with the branch office and community of readers staged on the magazine's front page than in the income from *Senki* sales. The 'masses' organized through the medium of *Senki* were regarded as material supporters or agents for the reconstruction of the JCP. However, it cannot be said that all readers were aware of how the yen they sent to *Senki* were used. Chōsen (Korea), a member of the 'community of readers,' was no exception.

Why would the KOPF publish a Hangul magazine, *Uri tongmu* (Our Comrades), at a time when reorganization was still underway? In 1928, the Fourth Congress of the Red International of Labor Unions (Profintern) adopted a thesis forcing colonial workers to join the trade union of their *country of residence*. In August of the same year, the Secretariat of the Comintern also reaffirmed the 'one country, one party' principle. In the interior of Japan, the dismantling of the Federation of Trade Unions of Zainichi Koreans in Japan began in earnest around September 1929, with the aforementioned Kim Tu-yong in overall charge of the task. In October 1931, the Japanese General Bureau of the Korean Communist Party and the Japanese headquarters of the Kōrai Communist Youth Association (a Korean youth group active during the Japanese occupation) published a joint statement in a Japanese Communist Party organ, the newspaper *Sekki* (Red Flag), and the KOPF became an organization under the Japanese Communist Party umbrella. The Chōsen Council was formed with KOPF and Dōshi-sha members such as Kim Tu-yong and Yi Puk-man as its core members. The instructions given to them by the KOPF were to "eliminate the national language of the imperialist state and make the mother tongue and national language the basis of creative practice" (5th Congress, 11–13 May 1932). As part of this policy, *Uri tongmu* was launched.

Let us now turn our attention to a Japanese-language advertisement in the April 1932 issue of the KOPF journal *Proletarian Culture*[28] which advertises the first issue of *Uri tongmu*.

> Dear all workers, peasants, and working masses across the country! *Uri tongmu*, a Korean-language magazine for all Korean workers who cannot read Japanese, will finally be launched in the coming May! […] In order to defend the publications of the Japan Federation of Proletarian Culture against the violent repression and interference of the ruling class and to protect *Uri tongmu* as your own, Japanese workers, peasants and Korean workers in Japan must become direct readers. Also, you should encourage as many Korean colleagues in your factories and workplaces to read this magazine as possible, so that the proletariat in Japan and Korea can form a revolutionary coalition. […] Even if it is only one or two sen, give generously. Please send funds to this magazine from your factories, workplaces, and farming villages. And please remember to order the magazine with advance payment, to protect the magazine.

> Let us grow *Uri tongmu* with a sense of class duty. Only then will it have a readership network like a spider's web throughout the nation's workplaces of Korean laborers, and the financial basis for publishing the magazine will be solidified. […] Let it rain 200 yen for the publication fund! [29]

If we replace the mention of the title *Uri tongmu* with *Senki* in the above quotation, we can see exactly the same structure as the advertisement examined in the previous section. Just as Korean readers who could not master the Japanese language as they wished sent funds to protect *Senki*, the ad urges "Japanese workers and peasants" who do not understand Korean to send funds to "become direct readers" and subscribe to the magazine ev-

ery month so that "the proletariat of Japan and Korea can form a revolutionary coalition," even if they do not understand the magazine's contents. The ad also emphasizes that a network of readers of *Uri tongmu* should be built in all workplaces of Korean laborers.

At this time, organizing or joining a network of readers of the KOPF's *Uri tongmu* or distributing the magazine was itself (even more than the network of readers of *Senki* and NAPF) considered an affiliated organization and source of funds for the JCP, and could be punished. The reason is that since the 1930s, such activities had been criminalized. As a result, the Tokkō began to investigate Korean residents in Japan. First, the Special Higher Police targeted the Zenkyō, which had targeted the Federation of Trade Unions of Zainichi Koreans in Japan for investigation. The new criteria for prosecution for ideological reasons, which allowed for prosecution simply if one's membership in an affiliated organization such as the Japan Proletarian Writers' League could be proved, gradually began to gain strength. In May 1931, the Supreme Court ruled that defendants could be punished if the police and prosecutors determined that they had contributed to the accomplishment of the Party's objectives.

Since the dissolution of the Federation of Trade Unions of Zainichi Koreans in Japan, the list of arrests of Zainichi Koreans by the Tokkō shows that the overwhelming majority of the prosecutions were for crimes of contribution to the accomplishment of the Party's objectives. The majority of those charged with this were suspected of distributing *Senki* and *Musansha* and organizing reader groups. In addition, many of them are affiliated with the Zenkyō. For example, Kim Mun-jun, who was indicted on 10 October 1930, was charged with the crime of accomplishment of purpose (distribution of *Mushin* newspaper, creation and distribution of leaflets), and his affiliation with "persons related to Zenkyō and the JCP" (*Tokkō geppō*, October 1930). Of course, those affiliated with *Senki* were also eventually charged with the same crime and indicted.[30]

In 1932, when *Uri tongmu* was first published, more than one-hundred KOPF officials were punished for association with it, because the KOPF was judged to be an offshoot of the Japanese Communist Party. The building of a readership network for *Uri tongmu* published by the KOPF, meant the creation of a "a readership network like a spider's web throughout the nation's workplaces of Korean laborers" was seen as a JCP auxiliary. Moreover, from the perspective of the KOPF, whose readership network was on the verge of collapse as the intensive crackdown by the Tokkō and the thought prosecutors was in full swing, the potential market of over 200,000 Korean readers must have been appealing. The mission of KOPF's *Uri tongmu* and the branch office and community of readers developed by the KOPF, including Yi Puk-man, was to bolster the agency of 'Korean malcontents' as it circulated among the 200,000 Koreans and the magazine was distributed. In the "Reports of the Chōsen Council" (*Puroretaria Bunka*, June/July 1933), criticism, reflections, and future initiatives concerning *Uri tongmu* are presented in detail.

The KOPF directs the collection of funds through deliveries and the organization of Koreans. The language "overcome the sectarian struggle of the Korean people" is also consistent with the policy of the JCP, which had emphasized that the struggle for colonial independence and class-based proletarian revolutionary struggle could not coexist. On the other hand, in June 1933, when this report was published, major conversions were covered by Japanese-language media, triggered by the conversion of Sano Manabu, who had served as chairman of the JCP ('conversion' refers to leftists renouncing their political beliefs, under pressure from Imperial authorities). As symbolized by the torture and murder of Kobayashi Takiji in February of the same year, the repression against the KOPF was becoming more severe by the day. In short, it is possible that the launch of KOPF's *Uri tongmu* was an alternative to overcome the limitations of the Japanese 'illegal commodity' market, which rapidly atrophied during this period. *Uri tongmu* should be reconsidered from the perspective of developing a new market for such 'illegal commodities'.

In the early 1930s, the reorganization of the Imperial publishing market was explored, and the colonial market was discovered. The April 1932 issue of the 'bourgeois magazine' *Kaizō* introduced literary writers from the colonies as its new commodity, including the

Korean Chō Kakuchū (Chang Hyŏk-chu), and the April 1932 issue of the left-wing magazine *Puroretaria Bunka* (Proletarian Culture) heralded the appearance of a new commodity called *Uri tongmu*. Also in the early 1930s, the claim emerged that Imperial colonies, never before consciously considered as independent publishing markets, would make possible "the realization of publishing imperialism". This text argues that the publishing world was being reformed by the fact that vast inventories of *enpon* "are inundating the colonies with a force that rivals the deployment of Japanese troops to Manchuria" (Minato 1931; Ko 2012).[31] At the same time that books produced by Imperial Japan's print capital flowed toward the colonial market as mobile media via rail, in Japan's interior, 'Korean malcontents' became 'mobile media,' and arrived at the colonial market in the interior bearing the new commodity *Uri tongmu*. In other words, *Uri tongmu* symbolizes the realignment of the Zainichi Korean bloc, which was once expected to be the foundation for the reconstruction of the Korean Communist Party, into an organization which served the reconstruction of the *Japanese* Communist Party. The market restructuring attempted by leftist print capital in Imperial Japan began at the intersection of two markets, empowered by the illusion of collective resistance to Imperial power, and the illusion that Koreans could become equal Imperial subjects if they entered a 'community of readers'.

Previous Japanese-language scholarship has tended to beautify cultural movements pursued by Japanese and Koreans around 1930 as solidarity among comrades–intellectual and political movements marked as sacred and above the predations of capital. As demonstrated in this essay, however, the publication and commodification of 'illegal' materials had a profound bearing on the movement of capital in the print industry. Furthermore, the struggle for survival waged in print by the Japanese socialist movement in the early 1930s resulted in a major blow to the organizational base and funding for the reconstruction of the Korean Communist Party, which was being advanced by Korean socialists in Japan. This particular chapter within Japan's cultural history of the late 1920s and early 1930s, the essay claims, should be subject to a degree of critical rethinking and rewriting.[32]

**Author Contributions:** Conceptualization: Y.R.K.; Conceptualization (partial-introduction; conclusion): K.-H.C.; Writing-drafting: Y.R.K.; Writing-review & editing: N.O. and K.-H.C.; Co-translation [from the Japanese original]: N.O.; Co-translation [from the Korean original]: K.-H.C.; Data-curation: Y.R.K. and N.O.; Methodology Y.R.K.; Investigation: Y.R.K.; Funding acquisition: K.-H.C. All authors have read and agreed to the published version of the manuscript.

**Funding:** This research received no external funding.

**Institutional Review Board Statement:** Not applicable.

**Informed Consent Statement:** Not applicable.

**Data Availability Statement:** Data sharing not applicable.

**Acknowledgments:** Young Ran Ko expresses her gratitude to Nick Ogonek and Kyeong-Hee Choi for translating the manuscript from Japanese and Korean into English, and for additional research to make the paper accessible to an Anglophone audience.

**Conflicts of Interest:** The authors declare no conflict of interest.

## Glossary

| | |
|---|---|
| All-Japan Federation of Proletarian Arts (NAPF\|Nippona Artista Proleta Federacio) | 全日本無産者芸術連盟 |
| Book Department (Home Ministry) | 図書課（内務省） |
| Bourgeois publishers | ブルジョワ出版社 |
| *Bungei sensen* (Literary Arts Front) | 『文芸戦線』 |
| Federation of Trade Unions of Zainichi Koreans in Japan | 在日朝鮮人労働総同盟 |

| | |
|---|---|
| Japan Federation of Proletarian Culture (KOFP \| Federacio de Proletaj Kultur-Organizoj Japanaj) | 日本プロレタリア文化連盟 |
| Japan Trade Union National Council (Zenkyō) | 日本労働組合全国協議会（全協） |
| Japanese Proletarian Writers' Series Publisher: Senki-sha (NAPF) | 日本プロレタリア作家叢書 |
| *Kaizō* (Reconstruction) Publisher: Kaizō-sha | 『改造』 |
| Left-wing (*sayoku*) publishers | 左翼出版社 |
| Leftish (*sayoku-teki*) publishers | 左翼的出版社 |
| Major leftish publishers: Kibōkaku, Dōjinsha, Kyōseikaku, Sōbunkaku, Marx Shobō, Iskura-kaku, Hakuyōsha, Sekaisha, Tetsutōshoin, Ueno Shoten, Nanban Shobō, Nansō Shoin, and Kōbundō Source: Kanroji Hachirō (1931) | 希望閣、同人社、共生閣、叢文閣、マルクス書房、イスクラ閣、白楊社、世界社、鐵塔書院、上野書店、南蠻書房、南宋書院、弘文堂 |
| Malcontent Koreans | 不逞鮮人 |
| *Musansha* (The Proletarian) | 『無産者』 |
| Publisher: Korean Federation of Proletarian Artists (KAPF \| Korea Artista Proleta Federacio) | 朝鮮プロレタリア芸術家同盟 |
| *Puroretaria geijutsu* (Proletarian Art) Publisher: Japan Proletarian Art League | プロレタリア芸術 |
| *Puroretaria Bunka* (Proletarian Culture) Publisher: KOPF | 『プロレタリア文化』 |
| Law of Temporary Control of Seditious Documents | 不穏文書臨時取締法 |
| *Senki* (Battle Flag) Publisher: Senki-sha (NAPF) | 『戦旗』/ NAPF |
| *Shuppan keisatsu hō* (Publication Police Reports) | 『出版警察報』 |
| *Tanemakuhito* (The Sower) | 『種蒔く人』 |
| Tokkō (Special Higher Police–full name: Tokubetsu Kōtō Keisatsu) | 特高（特別高等警察） |
| *Tokkō nippō* (Daily Special Higher Police Report on Publications) | 『特高日報』 |
| *Tokkō geppō* (Monthly Special Higher Police Report on Publications) | 『特高月報』 |
| *Uri tongmu* (Our Comrades) Publisher: KOPF | 『우리 동무』/KOPF |
| *Zen'ei* (Vanguard) Publisher: Union of Avant-garde Artists | 『前衛』 |

## Notes

1  The statistics are based on those reported in "*Senki's* Second Anniversary with May Day: Toppling Oppression and Surpassing the ¥300,000 Recruitment Fund" (Senki [May 1930], p. 61).

2  According to a Home Ministry Survey, around 1929 the circulation of *Kaizō* was 100,000, *Chūō kōron* 20,000, and *Bungei-shunjū* 70,000. Home Ministry *Shimbun zasshi to tsushinsha ni kan suru chō*, [Survey on Newspapers, Magazines and News Agencies] 1927; Reprinted in *Shimbun zasshi-sha toku-hi chōsa*, p. 21.

3  Discussions of the battle of "socialist commodity" between Kaizō-sha and Chuokōron sha, as well as of the complicitous relationship of *Senki* and *Bungei sensen* to commercial capital, also came up in Takashi Wada's (2022) article "Anti-Bourgeois Media

in the Japanese Proletarian Literary Movement" (*Humanities* 11, no. 6) which has much overlap with this paper. However, the content of this paper was based on two of my previous publications. See Ko "Teikoku Nihon no shuppan ichiba saihen to media ibento" and Ko "Shuppan teikoku no 'sensō'", pp. 127–37.

4  These translations follow a previous English translation by Christina Yi. See Ko "Censorship Empires, Illegal Commodities".

5  The edition produced by Kaizō-sha was issued 30 times (all 27 volumes) between June 1928 and October 1932. For more on the rivalry between the translation groups of the two editions, and the process by which 80 translators contributed to the Kaizō-sha edition and subsequently came into the limelight as Marxists, see Umeda, *Shakai undō to shuppan*, pp. 12–14.

6  Obi's *Shuppan to shakai* (Genshi Shobō, 2007) details the competitive advertising that developed in the Japanese-language media between the Kaizō-sha version and the leftist publisher group's version of the *Complete Works of Marx and Engels*. For more on the advertising for the Japanese-language edition, see Cheon, *Gendai no shodoku*, pp. 213–14.

7  Through a survey conducted by *Yomiuri* newspaper, the number of students who pre-ordered the books reached about 1000 at 25 national high school across the country. See *Yomiuri*, 6 July 1928.

8  See *Yomiuri*, 17 July 1928.

9  For information on Kobayashi Takiji at the time of writing "The Crab Cannery Ship," see Ogasawara, *Kobayashi Takiji*, p. 67; Shimamura, "Kobayashi Takiji '*Kanikōsen*' to chika katsudō-ka susu shakaishugi undo", p. 99. See also Field, *Kobayashi Takiji*, pp. 157–59.

10  See Toda, *Kanikōsen kesareta moji*, pp. 213–23 for an explanation of the publication background of the 1929 Senki-sha edition, and an appendix with a list of differences in *fuseji* between the Kaizō paperback edition (1933) and the Shinchō paperback edition (1933).

11  See Yamamoto, "Fuseji, ken'etsu, jiko-kisei. As for the relevant discussion of the history and function of *fuseji*, see Abel, *Redacted*, pp. 145–53 in particular, and Maki, *Fuseji no bunkashi*. Abel's book's seventh chapter (pp. 154–93) offers a detailed account of the use of *fuseji* in the magazine *Kaizō*. I also address the issue of colonialism in a larger study of modern and contemporary Japanese literature by taking up the role of *fuseji* and its mediatory role in Nakano Shigeharu's "Shinagawa Station in the Rain". See Ko, "'Ame no furu Shinagawa-eki' no *Musansha*-ban o tegakari toshi nagara senryaku to shite no Chōsen hyōshō", also included in Ko, *Sengo to iu ideorogī*, pp. 107–50.

12  See Ko, "Teikoku Nihon no shuppan ichiba saihen to media ibento", pp. 130–35.

13  The March 15 Incident (1928) and April 16 Incident (1929) were two days of mass arrests, under the newly revised and extremely broad Peace Preservation Law. Most of those arrested were individuals affiliated with organizations deemed "anti-government," largely communist and socialist groups.

14  See "Shakai undō no jōkyō: Fuhyō", pp. 1–53.

15  See Police Affairs Bureau, Home Ministry, ed., *Showa tairei keibi kiroku*, p. 177.

16  See Tsuboi, "Puroretaria zasshi no keiei", p. 192. In "The *Senki* Era" (*Minshū Hyōron* [April 1948]), pp. 49–50, Tsuboi writes that the most dangerous time for publishers was one just before binding was completed, but in such cases, publishers often divided the book binding process between two or three locations, taking measures so that if the process in one location was put to toll, those in the remaining locations could still accomplish the work safely. Originally, books were required to be deposited with the National Police Agency three days before they were issued, but in practice, deposition took place three days *after* release and issuance. Even if the issuance of the deposited book was prohibited, the publication had already been sold to some customers by the time the police went to the bookstore to seize it. For this reason, publishers of 'controversial' publications often pushed their readerships to purchase their items as early as possible.

17  "Zen-dokusha shokun ni uttafu" [An appeal to all readers], *Senki* September 1928, p. 157.

18  "Burujoa shuppan-butsu ni taisuru wareware no taido wa kaudenakereba naranu" [Our attitude toward bourgeois publications must be this way], *Senki* (June 1930), pp. 177–97.

19  For more on the founding and development of the magazine *Senki*, and the debates between Kurahara Korehito and Nakano Shigeharu, see Shockey, *The Typographic Imagination*, pp. 207–8.

20  Satō argues that "the fact that *Senki* published *Youth Senki* (No. 1–5) and *Homemaker's Senki* (only one issue) as separate volumes, and that it created about 300 reader branch organizations (as of September 1930) in factories and schools nationwide, were all counter-movements that directly challenged the mass mass-media nature of *Kingu*." Satō, "Kingu no jidai", p. 69. For more on *Youth Senki* and *Homemaker's Senki*, see Perry, *Recasting Red Culture in Proletarian Japan*.

21  See *Tokkō geppō* (March 1930), p. 73.

22  See (NAPF to Nihon Kyomintō to no kankei 1971, pp. 1015–16).

23  See *Tokkō Geppō* (April 1930): p. 66.

24  "Collecting for the second proletarian newspaper defense fund," "Let's protect the proletariat with our Might!" "Put seven *sen* of your return train fare toward the fund!" (*Senki* [January 1930]), p. 195.

25  "Sales of *Senki* have remained constant in recent years. Nevertheless, the ban on the release and distribution of has continued to be given to almost every issue, and [*Senki*] seems to be in a state of considerable financial distress. Its situation for the past month is as

follows: Income, 2500 yen (electricity fees, etc.)/Expenses: 1000 yen. Management expenses (labor, rent, etc.): 1200 yen. Printing costs 1500 yen. Paper (deficit), approximately 1200 yen. The company is currently 45,000 yen in debt for printing, paper, etc. The total amount of the *Senki* defense fund (announced in November) is 2328.19.5 yen". See (NAPF to Nihon Kyomintō to no kankei 1971, p. 1013).

26    See (NAPF to Nihon Kyomintō to no kankei 1971, p. 86).

27    300 yen out of a total income of 422 yen in August, 450 yen out of 719 yen in September, and 400 yen out of 959 yen in October (ibid., p. 89). Individual donations from *Senki* writers were also frequent. For example, Kobayashi Takiji was detained on June 24, 1930, on suspicion of being a sympathizer and contributor to the Communist Party for having donated the entire royalty from "The Crab Cannery Ship" to the party. The situation was the same for other writers, like Nakano Shigeharu (NAPF to Nihon Kyomintō to no kankei 1971, p. 152).

28    First published on May 1, the price was 5 *sen* plus postage. The magazine had 46 issues, 16 pages each, on onion-skin paper, and the purchase price had to be remitted to the KOPF publishing office.

29    *Puroretaria bunka,* (April 1932), pp. 82–83.

30    See Police Affairs Bureau, Home Ministry, ed., *Shōwa 5-nen chū ni okeru shakai undō no jōtai*, pp. 464–65.

31    See (Minato 1931, p. 123). The English translation of quotation is adapted from Ko "Censorship Empires, Illegal Commodities (tr. Christina Yi), pp. 127–34.

32    Previous versions of this paper were presented by Ko at conferences in Korea in 2008, 2010 and 2016, and published in the journal 사이間 *SAI* in Korean in 2009. Ko produced the original version in Korean and Japanese, based on original research. The English version by Ko, Ogonek and Choi is based on a shortened version of the Korean and Japanese, and includes additional research by all three authors to allow it to speak directly to Anglophone audiences. The longer version by Ko will appear as a chapter in her Japanese-language monograph to be published in 2023, tentatively titled *Shuppan tekikoku no sensō*.

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
