# Peer review of "Making Capital of ‘Illegal’ Publication under Japanese Imperial Censorship: Publication Strategies of Senki (Battle Flag) around 1930"

_humanities, doi:10.3390/h12050089_

Round 1
Reviewer 1 Report
This article is a strong contribution to the field of Japanese studies, providing some very detailed examples to the ways in which the publishing industry capitalized on the situation of mid-century publication bans. Despite displaying ample scholarly prowess with regard to the Japanese materials that relate to the topic, the article reveals stunning lack of reference to the myriad works of English language scholarship on the topic. Disregarded are not only the content of the work of J Rubin, GJ Kazsa, GT Shea, N Field, S Perry, J Abel, N Shockey, R Mitchell, E Tipton, , but also their findings. This makes the article either repeat some of their findings without citation and presented as though new and original or simply not clear on its own contribution to the field. Particularly shocking in this regard are the bits that refer to Nakano's Shinagawa Eki poem which has been much discussed. In addition the recent work of J. Abel and Y. Maki on the subject of fuseji is not referred to at all. The main argument (that bans can be used profitably) is also one made in much of the more theoretical scholarship on the subject of censorship both within and beyond Japanese studies, though none of this scholarship is mentioned and the particular contribution of this article to those discussions goes unmentioned. For this paper to be publishable it must engage in these discussions.
There are also some more specific things that should be attended to before publication.
The author does not give adequate justification for using the translations of other scholars rather than simply providing their own translations.
"Leftish publishers" is an awkward translation into English. Depending on the original Japanese (poi? or teki? or kei? or something else?), perhaps something generic like "left-related" or "left-leaning" or simply "leftist" would be less obtrusive in English. Footnote 12 does not resolve this issue satisfactorily enough.
missing word on page 7. Should read "relied heavily ON publishing work"
On page 12, the "the" in "the greatest cause of THE Senki's evolution" should be deleted.
On 14-15, "the yen they sent to Senki was used" should read "the yen they sent to Senki were used"
Reviewer 2 Report
This is a well-researched and highly informative paper that examines the underexplored publication strategies among proletarian cultural movements to evade censorship and compete with mainstream media in imperial Japan. It does so by investigating the prominent proletarian cultural journal Senki (Battle Flag) and how its editors navigated an increasing oppressive publication landscape. The paper focuses on legal and illegal (banned) leftist publications and how those involved in the publication of Senki aimed to benefit from the label “illegal” by making it part of their promotion and marketing. Of particular interest is how the publication strategies of proletarian cultural movements competed with mainstream and profit-driven publishers. Altogether, the paper makes an important contribution to the field of modern Japanese literature as well as studies on social movements and censorship.
Suggestions:
1) A weakness of the paper is its narrow focus, making it difficult to cater to a wider readership. It hardly engages with Anglophone scholarship on the topic. A broader engagement with recent scholarship on Japan’s publishing industry and censorship during the interwar period would help to better articulate and present the paper’s contribution and with what research field(s) it engages.
2) In relation to the paper’s narrow focus, a more substantial historization and contextualization would strengthen the paper’s claims and better inform readers outside of Japanese studies, who are not familiar Japanese history and (proletarian) literature.
3) A clearer conceptualization of print capitalism and mechanical reproduction informed by, for example, media theory and Marxist theory would strengthen the paper’s argument to demonstrate what is at stake for proletarian cultural movements and illustrate the complex nature of the publishing market under censorship in Imperial Japan.
4) The ambivalent relationship between print capitalism, media, and leftist movements needs to be better addressed. This is to say that the history of an arguably ideological struggle/contradiction between leftist thought and print capitalism is not only limited to proletarian cultural movements interwar Japan, but also to earlier leftist movements in Japan and elsewhere, such as anarchism. Leftist movements are made possible and are a product of print capitalism as well as the imperial/capital infrastructure and technology in the first place. In that case, how do we assess the conflict between ideology, political activism, and print capitalism?
5) How does the usage of “illegal” as a marketing strategy by proletarian cultural movements in their publications, such as Senki, relate to other marketing strategies such as the “prison experience” (Soeda, 2016) or the “tenkō experience” among proletarian writers? Were the capital driven attempts among proletarian cultural movements different or not from “bourgeois” media? To what extent did proletarian publishers focus on profit and exploit workers to produce its “commodities”? The paper shows that a lot of the income/revenue was used for other proletarian projects, which seems to be different from “bourgeois” media. Also, what about different publication methods and strategies between large proletarian cultural movements like NAPF and KOPF and smaller, regional proletarian coteries? What other methods did NAPF and KOPF have to raise money to fund their activities?
Minor typos/remarks:
*Some of the wording in the abstract is not entirely clear and easy to understand.
* The repetitive use of “commodity.” Perhaps alternate with “publications” “books” “journals”
* The conclusion is somewhat short and abrupt. It would benefit from a clearer summary of the paper and its argument.
*p.6 note 13. Shimamura Haru = Shimamura Teru?
*p.6 In the text, the author notes that Kanroji “rigid distinction is not entirely accurate” but on p.6 note 12, it is stated that the “essay follows the distinction between left-wing and leftish in accordance with Kanrōji’s understanding.” Also, see spelling of Kanroji / Kanrōji.
p. 11 last sentence of section 3 “…collection of such yen – of currency.” Typo?
p. 12 “…and its offices were was used…”. Typo.
p. 14 last sentence “…how the yen they sent to…” Typo?
p. 15 reading for赤旗. For the prewar newspaper publication, I believe the pronunciation Sekki is often used. Please check.
p. 15 the quotation is not fully indented, making it difficult for the reader to recognize the section as quotation.
Reviewer 3 Report
This is an excellent, well-written piece that explores an angle of the Proletarian Literature and Arts Movement that few scholars have written about: the commodification of "illegal" publications during late 1920s and early 1930s imperial Japan when the state began to throw money at tamping down leftism. I enjoyed the discussion of "swag" and "merch" that Senki brought out to finance its activities, such as a bust of Lenin and other goodies--one wonders if these items can still be found in Tokyo flea markets or high-end Jimbocho bookstores! I am reminded of Verso Books' own commodification of popular critiques of capitalism and even "limited edition" canvas bag advertising a protest slogan and the bookseller. Fascinatingly, every time Senki had a banned edition, the author points out that this actually increased circulation numbers, since readers would subscribe and directly buy copies before they were seized by the Tokko. (I can imagine a black market of enterprising Tokko officials secretly selling the seized and allegedly destroyed copies to used booksellers on the sly--was there a such thing?) Nowadays, with various currents of what we might call "fascism" on the rise globally, there correspondingly seems to be a growth of interest in Communism, leftwing thought, and other writings inaccessible during Cold War times. It is also fascinating that overseas Koreans then became a new target market for the publications of leftwing publishers, and interesting that censorship was less heavy here. Was this because those loyal to imperial Japan's mission to atrophy any challenges to the Emperor system were not able to read Korean? I assume that it was not easy to hire ethnic Koreans to scour writings in publications for potential spurious content? Anyway, this article was a joy to read, and a very timely topic! It also amazes me how clever business tactics in Japan, used even by those highly critical of capitalism, have assisted in maintaining a vibrant commercial culture in Tokyo. When can we finally buy a bust of Kobayashi Takiji for our desks? I'm also amazed at the more mainstream fertilization of wide-readership magazines like Kaizo and Chuo Koron with so-called proletarian literature, and how popular it was with the bourgeoisie.
Some small points. Recent literature by Japanese scholars and some Western scholars is well-cited. Work by Heather Bowen-Stryk and Mika Endo, for example, would be useful to include in your references. How about the works of Korean researchers on the topic?
